

# Retrospective screening of serum IgG glycosylation biomarker for primary Sjögren's syndrome using lectin microarray

Siting Li[1,*], Xiaoli Zeng[1,2,*], Shiyi Tang[1], Xi Li[1,3], Guoyuan Zhang[4], Mengtao Li[1], Xiaofeng Zeng[1] and Chaojun Hu[1]

[1] Department of Rheumatology, Peking Union Medical College Hospital, Peking Union Medical College & Chinese Academy of Medical Sciences, National Clinical Research Center for Dermatologic and Immunologic Diseases (NCRC-DID), Key Laboratory of Rheumatology & Clinical Immunology, Ministry of Education, Beijing, China
[2] Affiliated Hospital of North Sichuan Medical College, Nanchong, China
[3] Department of Clinical Laboratory, First Affiliated Hospital of Guangxi Medical University, Nanning, Guangxi, China
[4] Department of Laboratory Medicine, North Sichuan Medical College, Nanchong, China
* These authors contributed equally to this work.

Corresponding authors
Chaojun Hu, huchaojun818@qq.com
Xiaofeng Zeng,
zengxfpumc@163.com

## ABSTRACT

**Background:** Primary Sjögren's syndrome (PSS) is a systemic autoimmune disease resulting in significant loss of systemic gland secretory function. IgG glycosylation abnormalities had been found to play important roles in autoimmune diseases. Here, we aim to explore the specific changes of IgG glycosylation in PSS patient serum that could serve as potential biomarkers for disease diagnosis and differential diagnosis.
**Method:** From 2012 to 2018, patients diagnosed with PSS or primary biliary cholangitis (PBC) admitted consecutively to the department of Rheumatology at Peking Union Medical College Hospital were retrospectively included in this study. Glycan profiles of serum IgG from 40 PSS patients, 50 PBC patients, and 38 healthy controls were detected with lectin microarray containing 56 lectins. Lectins with significantly different signal intensity among groups were selected and validated by lectin blot assay.
**Results:** Lectin microarray analysis revealed that binding levels of Amaranthus Caudatus Lectin (ACL, prefers glycan Galβ3GalNAc, $P = 0.011$), Morniga M Lectin (MNA-M, prefers glycan mannose. $P = 0.013$), and Lens Culinaris Agglutinin (LCA, prefers glycan fucose) were significantly increased, while *Salvia sclarea* Agglutinin (SSA, prefers glycan sialylation, $P = 0.001$) was significantly decreased in PSS patients compared to PBC group. Compared to healthy controls, MNA-M ($P = 0.001$) and LCA ($P = 0.028$) were also significantly increased, while Phaseolus Vulgaris Erythroagglutinin and Phaseolus Vulgaris Leucoagglutinin (PHA-E and PHA-L, prefer glycan galactose, $P = 0.004$ and $0.006$) were significantly decreased in PSS patients. The results of LCA and MNA-M were further confirmed using lectin blot assay.
**Conclusion:** Changes in serum IgG glycosylation in PSS increased binding levels of LCA and MNA-M lectins using microarray techniques compared to PBC patients and healthy controls, which could provide potential diagnostic value. Increased core fucose and mannose alteration of IgG may play important roles in PSS disease.

## INTRODUCTION

Primary Sjögren's syndrome (PSS) is a complex heterogeneous autoimmune disease characterized by lymphocytic infiltration of the secretory glands and significant loss of secretory function with oral and eye dryness, as well as extra-glandular involvement that may impair the musculoskeletal, pulmonary, renal, neurological, and other organs/systems (*Both et al., 2017*; *Ramos-Casals et al., 2012*). PSS is the second most common connective tissue disease after rheumatoid arthritis and affects predominantly middle-aged women with a female/male incidence of approximately 9:1 (*Beckman et al., 2017*; *Qin et al., 2015*). Although PSS is currently not yet fully understood, increased activation of B cells and autoantibody production, such as anti-SSA/Ro and anti-SSB/La autoantibodies, are thought to play important roles. As standard diagnostic biomarkers, the presence of anti-SSA and anti-SSB were only 52–67% and 49% in PSS respectively (*Liu et al., 2017*). Due to its non-specific symptoms, PSS is sometimes difficult to recognize, and diagnosis may be delayed by more than 10 years (*Parisis et al., 2020*; *Witte, 2019*). Primary biliary cirrhosis (PBC) is a complex systemic disease characterized by chronic non-suppurative destructive cholangitis and is most often overlapped with Sjögren's syndrome (SS) (*Gershwin et al., 2005*; *Watt, James & Jones, 2004*). These coexisting conditions frequently make it more difficult the diagnosis and treatment of the disease.

Glycosylation is the most complex post-translational modification of proteins and has profound structural and functional effects on the conjugate (*Eichler, 2019*). It is estimated that more than half of human proteins are glycosylated with different glycan chains (*Christiansen et al., 2014*).

Immunoglobulin G (IgG) IgG is mostly N-glycosylated in the heavy constant region. To date, numerous studies have confirmed that changes in IgG glycosylation have important roles in the regulation of effector functions (*Dekkers et al., 2017*; *Quast, Peschke & Lünemann, 2017*; *Wang, 2019*). For instance, a lack of core fucose leads to enhanced antibody-dependent cell-mediated cytotoxicity (ADCC) activity. Aberrant IgG glycosylation has been found in various autoimmune diseases such as rheumatoid arthritis (RA), systemic lupus erythematosus (SLE), and inflammatory bowel disease (IBD) (*Bondt et al., 2013*; *Shinzaki et al., 2013*; *Sjöwall et al., 2015*). Therefore, the structural analysis of glycans in IgG is critical in understanding respective autoimmune diseases. However, little has been reported on the IgG glycosylation profile for PSS.

Lectin microarray is an emerging technology for the study of glycosylation (*Hirabayashi et al., 2013*). Compared with conventional glycan analysis methods such as mass spectrometry, it provides simple procedures for differential complex glycan profiling in a rapid, high-throughput, and high-sensitivity manner, and does not require prior liberation of glycans from the core protein which may destroy their native structure (*Hirabayashi, 2014*; *Hirabayashi, Kuno & Tateno, 2015*). Lectin microarray has already found maximum

use in diverse fields of glycobiology and made remarkable achievements in the study of glycosylation and biomarker identification for tumors and autoimmune diseases (*Dang et al., 2020*; *Hashim, Jayapalan & Lee, 2017*; *Li et al., 2019*). In this study, we used lectin microarray for the first time to analyze the expression profile of serum IgG glycosylation in patients with PSS, PBC, and healthy controls. Lectin blot was performed to validate the differences and obtain oligosaccharides specifically expressed in PSS.

## METHODS

### Patients and samples

Patients diagnosed with PSS or PBC admitted consecutively to the department of Rheumatology at Peking Union Medical College Hospital during the period from 2012 to 2018 were retrospectively included in this study. PSS was diagnosed according to the 2012 American College of Rheumatology (ACR) criteria (*Shiboski et al., 2012*), and PBC was diagnosed according to the American Association for the Study of Liver Diseases criteria (*Heathcote, 2000*). Patients meeting the classification criteria of more than one autoimmune disease or with cancer were excluded. A total of 128 serum samples were used for lectin microarray analysis, obtained from 40 PSS patients, 50 PBC patients, and 38 healthy controls (HCs) who were healthy volunteers without autoimmune diseases. In addition, to verify the significant findings, we randomly selected 12 PSS patients, 12 PBC patients, and 12 HCs from the microarray cohort, and combined them with a new cohort of serum samples including 16 PSS patients, 16 PBC patients, and 16 HCs for lectin blot analysis. Serum samples were collected upon admission, allowed to clot at room temperature for 30 min, centrifuged for 5 min at 1,000×*g*, and stored at −80 °C until used. Autoantibodies were tested using chemiluminescence immunoassay (YHLO Biotech Co., Shenzhen, China). The study was conducted in accordance with the Declaration of Helsinki, and approved by the Ethic Committee of Peking Union Medical College Hospital (Approval Code: S-478 Approval Date: 2012-10-31). All subjects gave written informed consent.

### Lectin microarray

Totally 128 serum samples were detected using a commercial lectin microarray (BCBIO Biotech, Guangzhou, China) with 56 lectins, which had been proved of its reliability and used in biomarker finding previously (*Li et al., 2021*; *Sun et al., 2016*). The detailed glycan binding specificities and type of linkage for each lectin could be found in Supplemental File. Detailed procedure could be seen in our previous works (*Hu et al., 2020*; *Li et al., 2022a*; *Li et al., 2022b*; *Zeng et al., 2021*). Briefly, lectin microarrays were taken out from −80 °C and warmed up at room temperature for half an hour, then they were incubated with a blocking buffer (3% BSA in PBS) at room temperature for 2 h. After washing three times with PBS, 200 µl of 1:1,000 diluted samples serum was added and incubated with the microarrays at 4 °C overnight. The microarrays were washed three times with PBS and then incubated with 5 mL of 1:1,000 diluted Cy5-labeled goat anti-human IgG antibody (Jackson Laboratory, Bar Harbor, ME, USA) in the dark at room temperature for 50 min. Finally, after three PBS washes, microarrays were rinsed with distilled water and dried.

Microarrays were scanned with the GenePix 4000B Microarray Scanner (Molecular Devices, Sunnyvale, CA, USA).

## Lectin microarray data analysis

For lectin array assays, the median foreground and background fluorescent intensity for each spot on the arrays were acquired using the GenePix Pro 6.0 software (*Li et al., 2022b*). We calculated the signal-to-noise ratio (S/N) (the medium intensity of the spot foreground relative to the background) of each lectin spot. To prevent bias of the lectin microarray from the inter-array, we normalized the S/N data in terms of controls between arrays (*Silver, Ritchie & Smyth, 2009*). The following rules according to the method of *Hu et al. (2021)* were used to identify significant differences in the binding activity of lectins between subject groups: (a) fold change (group1 (S/N)/group2 (S/N)) $\geq 1.3$ or $< 0.77$, (b) *P*-value $< 0.05$.

## Lectin blot

To validate the results of the differences in lectin microarray analysis, lectin blot was used to detect serum samples which were collected from 12 PSS patients, 12 PBC patients, and 12 HCs randomly selected from the lectin microarray analysis cohort, and 16 PSS patients, 16 PBC patients, and 16 HCs from a new cohort.

First, serum samples were diluted by $1 \times$ PBS, mixed with gel electrophoresis loading buffer (CWbiotech, Beijing, China) to a final 1:100 ratio, and boiled for 10 min. Twenty microliters per sample were separated by 10% sodium dodecyl sulphate–polyacrylamide gel electrophoresis (SDS–PAGE) and electrotransferred onto polyvinylidene fluoride membranes (Millipore, Billerica, MA, USA) (*Li et al., 2022b*). After washing two times with PBS Tween, the membrane was incubated with 10× Carbo-Free Blocking Solution (1:10; Vector Laboratories Inc., Newark, CA, USA) at room temperature for 2 h. Then, the membranes were washed twice and incubated with 20 µg/mL of Cy3-labeled (1:1,000; GE Healthcare, Chicago, IL, USA) LCA and MNA-M lectins at 4 °C overnight in the dark. Finally, the washed and dried membranes were detected by a fluorescence signal system of Typhoon FLA 9500 (GE Healthcare, Chicago, IL, USA).

## Statistical analysis

SPSS 22.0 was used to perform all statistical analyses and GraphPad Prism 8 was used to draw plots in the study (*Zeng et al., 2021*). Continuous variables were expressed as mean ± standard deviation. The differences among the PSS, PBC, and HC groups were tested by one-way analysis of variance (ANOVA) with Tukey's HSD test. *P*-value less than 0.05 was considered statistically significant.

# RESULTS

## Patient characteristics

As listed in Table 1, a total of 128 serum samples were used for lectin microarray analysis, obtained from 40 PSS patients (48.52 ± 9.73 years of age; 36 females), 50 PBC patients (52.30 ± 10.13 years of age; 46 females), and 38 healthy controls who were healthy

**Table 1 Clinical and laboratory characteristics of all 212 subjects.**

| Parameter | Lectin microarray | | | Lectin blot | | |
|---|---|---|---|---|---|---|
| | PSS (*n* = 40) | PBC (*n* = 50) | HC (*n* = 38) | PSS (*n* = 28) | PBC (*n* = 28) | HC (*n* = 28) |
| Sex (M/F) | 4/36 | 4/46 | 3/35 | 4/24 | 1/27 | 3/25 |
| Age (y) | 48.52 ± 9.73 | 52.30 ± 10.13 | 45.60 ± 7.64 | 46.39 ± 10.05 | 52.21 ± 11.92 | 41.14 ± 6.76 |
| Laboratory results | | | | | | |
| Anti-SSA+ (*n*, %) | 38 (95.0%) | 4 (8.0%) | 0 | 27 (96.4%) | 1 (3.6%) | 0 |
| Anti-SSB+ (*n*, %) | 29 (72.5%) | 1 (2.0%) | 0 | 14 (50.0%) | 1 (3.6%) | 0 |
| AMA-M2+ (*n*, %) | NA | 37 (74.0%) | 0 | NA | 23 (82.1%) | 0 |
| Anti-dsDSA+ (*n*, %) | 6 (15.0%) | 0 | 0 | 3 (10.7%) | 0 | 0 |
| Anti-Scl-70+ (*n*, %) | 1 (2.5%) | 2 (4.0%) | 0 | 1 (3.6%) | 1 (3.6%) | 0 |

Note:
PSS, Primary Sjögren's syndrome; PBC, primary biliary cholangitis; HC, health control; NA, not available. Lectin blot samples were collected from 12 PSS patients, 12 PBC patients, and 12 HCs randomly selected from the lectin microarray analysis cohort, and 16 PSS patients, 16 PBC patients, and 16 HCs from a new cohort.

**Table 2 Significant differences in binding between IgG and lectin in PSS, PBC, and HCs.**

| Lectin | Normalized fluorescence intensity (Mean ± SD) | | | Fold change | | | |
|---|---|---|---|---|---|---|---|
| | PSS | PBC | HC | PSS/PBC | *P* | PSS/HC | *P* |
| SSA | 2.57 ± 1.64 | 3.57 ± 1.86 | 2.60 ± 1.16 | 0.72 | 0.001** | 0.99 | 0.934 |
| LCA | 5.00 ± 3.10 | 3.71 ± 1.45 | 3.75 ± 1.56 | 1.35 | 0.011* | 1.33 | 0.028* |
| MNA-M | 5.79 ± 3.88 | 4.20 ± 1.92 | 3.68 ± 1.21 | 1.37 | 0.013* | 1.57 | 0.001** |
| ACL | 2.03 ± 1.36 | 1.48 ± 0.61 | 2.40 ± 3.06 | 1.37 | 0.012* | 0.85 | 0.490 |
| PHA-E | 7.44 ± 4.32 | 8.34 ± 4.41 | 9.81 ± 4.50 | 0.89 | 0.334 | 0.75 | 0.004** |
| PHA-L | 8.32 ± 4.99 | 9.81 ± 5.87 | 11.12 ± 5.36 | 0.85 | 0.206 | 0.75 | 0.006** |

Notes:
** $P < 0.01$.
* $P < 0.05$.
PSS, Primary Sjögren's syndrome; PBC, primary biliary cholangitis; HC, health control; SSA, *Salvia sclarea*; LCA, Lens Culinaris Agglutinin; MNA-M, Morniga M Lectin; ACL, *Amaranthus caudatus* lectin; PHA-E, *Phaseolus vulgaris* Erythroagglutinin; PHA-L, *Phaseolus vulgaris* Leucoagglutinin.

volunteers (45.60 ± 7.64 years of age; 35 females). A set of 12 PSS patients, 12 PBC patients, and 12 HCs randomly selected from lectin microarray analysis together with a new cohort of samples (including 16 PSS patients (43.44 ± 9.58 years of age; 13 females), 16 PBC patients (53.75 ± 12.75 years of age; 15 females), and 16 health controls (35.19 ± 5.06 years of age; 15 females)) was collected to verify significant findings using lectin blot. Autoantibody tests indicated that anti-SSA positivity was observed in 95% and 96.4% of the microarray and lectin blot PSS cohorts, while anti-SSB positivity was observed in 72.5% and 50%, respectively. AMA-M2 positivity was present in 74% and 82.1% of the microarray and lectin blot PBC cohorts.

### Lectin microarray analysis for serum IgG glycosylation

Overall results of 56 lectins were presented in Fig. S1. Significant results of the lectin microarray were shown in Table 2. Compared to HCs, binding levels of MNA-M (prefers glycan mannose, fold change 1.57, *P* = 0.001) and LCA (prefers glycan fucose, fold change

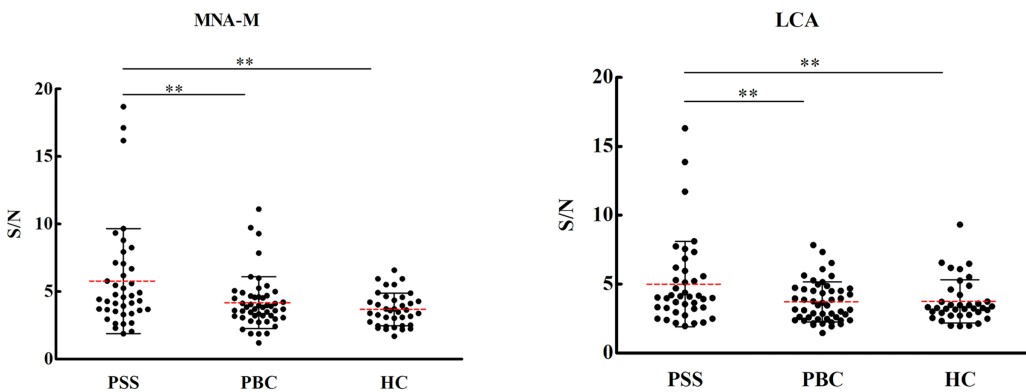

**Figure 1 Specific changes of serum IgG glycosylation from the lectin microarray.** $^{**}P < 0.01$. Red bars represent the mean ± standard deviation. PSS: Primary Sjögren's syndrome; PBC: Primary biliary cho-langitis; HC: Health control; LCA, Lens Culinaris Agglutinin; MNA-M, Morniga M Lectin; S/N, the medium intensity of the spot foreground relative to the background.

1.33, $P = 0.028$) were increased, while PHA-E and PHA-L (prefer glycan galactose, fold change 0.334 and 0.206, $P = 0.004$ and 0.006) were decreased in PSS patients. Compared to PBC patients, the signal intensities of the lectins MNA-M (fold change 1.37, $P = 0.013$), LCA (fold change 1.35, $P = 0.011$), and ACL (prefers glycan Galβ3GalNAc, fold change 1.37, $P = 0.012$) were significantly increased, while that of lectin SSA (prefers glycan sialylation, fold change 0.72, $P = 0.001$) was significantly decreased in serum IgG from PSS patients. As demonstrated in Fig. 1, PSS patients' serum IgG had significantly higher affinities for MNA-M and LCA in comparison with PBC and health controls ($P < 0.01$). Receiver operating characteristic (ROC) analysis revealed area under the curve (AUC) levels of 0.635 and 0.660 for MAN-M and LCA. The fold-change results of all lectins for PSS compared to PBC and HC were illustrated in Figs. S2 and S3.

### Lectin blot analysis

Since significant differences were observed only for MNA-M and LCA among PSS, PBC, and HC groups, the two lectins were selected to validate the microarray results. MNA-M results showed that PSS patients had a higher affinity for MNA-M in comparison with PBC patients and HCs, indicating an increased binding level of mannose in serum IgG from patients with PSS (Flurorescense intensity signal PSS: $90.72 * 10^3 ± 23.85 * 10^3$, PBC: $69.93 * 10^3 ± 138.45 * 10^3$, HC: $71.49 * 10^3 ± 126.56 * 10^3$, $P < 0.01$, Fig. 2), which was consistent with the result from the lectin microarray. LCA results showed that PSS patients had a higher affinity for LCA compared to PBC patients and HCs, indicating an increased binding level of fucose in serum IgG from patients with PSS (flurorescense intensity signal PSS: $122.04 * 10^3 ± 42.51 * 10^3$, PBC: $84.64 * 10^3 ± 33.67 * 10^3$, HC: $71.06 * 10^3 ± 25.59 * 10^3$, $P < 0.01$, Fig. 3), which was also consistent with the results from the lectin microarray.

### DISCUSSION

Numerous studies have confirmed that the change of IgG Fc glycosylation has an important effect on the activity of antibodies (*Wang, 2019*; *Wang & Ravetch, 2019*), and

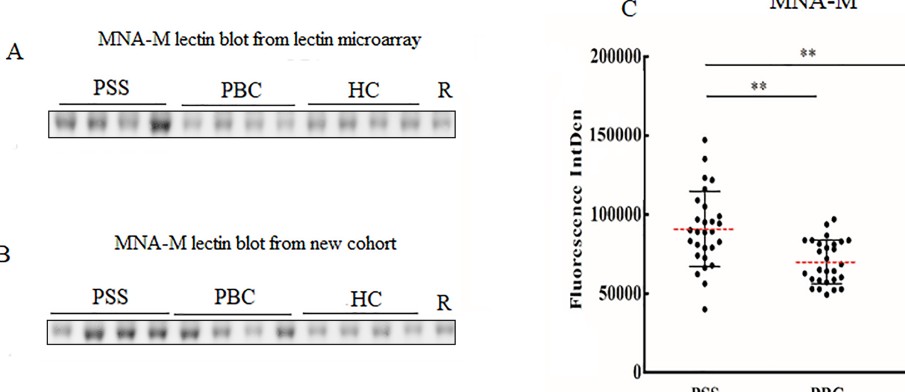

**Figure 2 Lectin blot of MNA-M lectin for serum IgG.** (A) Lectin blot of MNA-M for serum IgG selected from Lectin microarray cohort. (B) Lectin blot of MNA-M for serum IgG selected from a new cohort. (C) Specific changes of MNA-M lectin blot bands combining (A) and (B). **$P < 0.01$. Red bars represent the mean ± standard deviation. PSS, Primary Sjögren's syndrome; PBC, Primary biliary cholangitis; HC, Health controls; MNA-M, Morniga M Lectin; R, Reference.

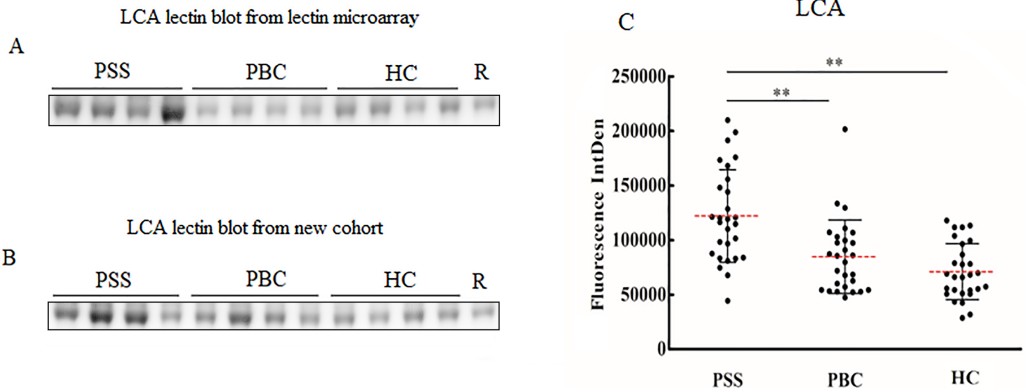

**Figure 3 Lectin blot of LCA lectin for serum IgG in PSS patients.** (A) Lectin blot of LCA for serum IgG selected from Lectin microarray cohort. (B) Lectin blot of LCA for serum IgG selected from a new cohort. (C) Specific changes of LCA lectin blot bands combining (A) and (B). **$P < 0.01$. Red bars represent the mean ± standard deviation. PSS, Primary Sjögren's syndrome; PBC, Primary biliary cholangitis; HC, Health control; LCA, Lens Culinaris Agglutinin; R, Reference.

has an important role in the occurrence and development of autoimmune diseases (*Li et al., 2019*; *Seeling, Brückner & Nimmerjahn, 2017*). Lectin microarray is based on the interaction of lectins with glycans and enables rapid, high-throughput, and high-sensitivity profiling of complex glycans features. In this study, lectin microarray with 56 kinds of lectin was used to detect the structures of serum IgG oligosaccharides in patients with PSS. Elevated binding of mannose and fucose was specifically observed in PSS patients compared to PBC patients and HCs. Binding of Galβ3GalNAc was increased and sialyation was decreased for PSS compared to PBC patients, while binding of galactosylation was decreased compared to healthy controls.

Regarding distinguishing PSS from healthy controls, a lack of the core fucose of IgG can significantly increase its affinity for the FcγRIIIa receptor and promote ADCC (*Bruggeman et al., 2017*; *Ferrara et al., 2011*). Bisecting GlcNAc was also associated with a decrease in core fucose. Concerning mannose glycan, a recent study using the same lectin microarray found that an elevated level of IgG4 mannose was associated with lacrimal and salivary glands' involvement in IgG4-related disease, possible through the complement lectin pathway (*Hu et al., 2021*). Additionally, similar to our result, previous research has also reported that PSS expressing rheumatoid factor (RF) exhibited low expression of galactose in serum IgG (*Bond et al., 1996*; *Kuroda et al., 2002*). Galactose is the most variable IgG glycosylation trait at the population level (*Krištić et al., 2014*) and can change quickly in acute inflammation (*Novokmet et al., 2014*). Fc galactosylation is necessary for the efficient initiation of the anti-inflammatory signaling cascade through binding to the inhibitory receptor FcγRIIb (*Reily et al., 2019*). IgG agalactosylated structures (IgG-G0) were significantly increased in patients with RA and positively correlated with disease activity (*Gińdzieńska-Sieśkiewicz et al., 2016*). Combined with our results, we speculated that the altered IgG glycosylation patterns might contribute to the pathogenesis of PSS such as secretory gland destruction.

Compared to PBC patients, our study indicated that binding levels of glycan mannose, fucose, and Galβ3GalNAc were increased, while that of sialylation was decreased in PSS. Galβ3GalNAc is the core 1 structure of O-glycosylation, which has only been observed in the hinge region for IgG3. Though not fully investigated, this alteration might prevent the immunoglobulin from proteolytic degradation and assist in antigen-binding of the Fab fragment by maintaining flexibility (*Plomp et al., 2015*). Sialylation of the IgG Fc domain has been found to negatively regulate the complement-dependent cytotoxicity (CDC) effect (*Quast et al., 2015*). An abnormally high level of asialylated IgG had also been observed in the previous study (*Basset et al., 1998*). In all, unique IgG glycosylation patterns of PSS and PBC may provide a reasonable direction for identifying PSS from PBC.

Apart from specific glycans, our study also suggested that high-throughput lectin microarray was a convenient and robust method for studying glycosylation for autoimmune diseases. Although levels of specific glycans could not be quantitatively determined, different levels of glycan-binding would still be valuable for disease diagnosis and differential diagnosis. Alteration of affinity for lectins MNA-M and LCA could serve as specific disease biomarkers for PSS patients and provide additive value in diagnosis.

Our study has some limitations. Due to a restricted number of clinical samples, autoimmune disease controls apart from PBC and SLE were not included. Baseline information and laboratory results were not sufficiently collected for the patients. The cohorts should also be expanded to validate the finding. Although lectin microarray could serve as a convenient tool for glycosylation study, the exact structure and site of glycosylation could not be clarified. Since serum IgG types and levels were not adjusted for the analysis, changes in lectin binding may not fully reflect the degree of specific glycan. In the future, other techniques such as MS would be combined to further investigate the role of glycosylation in PSS.

## CONCLUSION

Changes in serum IgG glycosylation in PSS increased binding levels of LCA and MNA-M lectins compared to healthy controls and PBC patients using microarray techniques, which could provide potential diagnostic value. Elevated levels of fucose and mannose may play important roles in the development of PSS.

### Funding

This study was supported by the National High Level Hospital Clinical Research Funding (2022-PUMCH-A-039, 2022-PUMCH-B-013), the  National Key Research and Development Program of China (2019YFC0840603, 2017YFC0907601, and 2017YFC0907602), the National Natural Science Foundation of China (81771780), and the CAMS Initiative for Innovative Medicine (2017-I2M-3-001 and 2019-I2M-2-008). The funders had no role in study design, data collection and analysis, decision to publish, or preparation of the manuscript.

### Grant Disclosures

The following grant information was disclosed by the authors:
National High Level Hospital Clinical Research Funding: 2022-PUMCH-A-039, 2022-PUMCH-B-013.
National Key Research and Development Program of China: 2019YFC0840603, 2017YFC0907601, and 2017YFC0907602.
National Natural Science Foundation of China: 81771780.
CAMS Initiative for Innovative Medicine: 2017-I2M-3-001 and 2019-I2M-2-008.

### Competing Interests

The authors declare that they have no competing interests.

### Author Contributions

- Siting Li conceived and designed the experiments, performed the experiments, analyzed the data, prepared figures and/or tables, authored or reviewed drafts of the article, and approved the final draft.
- Xiaoli Zeng conceived and designed the experiments, performed the experiments, analyzed the data, prepared figures and/or tables, authored or reviewed drafts of the article, and approved the final draft.
- Shiyi Tang performed the experiments, prepared figures and/or tables, and approved the final draft.
- Xi Li performed the experiments, prepared figures and/or tables, and approved the final draft.
- Guoyuan Zhang conceived and designed the experiments, authored or reviewed drafts of the article, and approved the final draft.

- Mengtao Li conceived and designed the experiments, authored or reviewed drafts of the article, and approved the final draft.
- Xiaofeng Zeng conceived and designed the experiments, analyzed the data, authored or reviewed drafts of the article, and approved the final draft.
- Chaojun Hu conceived and designed the experiments, analyzed the data, authored or reviewed drafts of the article, and approved the final draft.

## Human Ethics

The following information was supplied relating to ethical approvals (*i.e.*, approving body and any reference numbers):

The study was conducted in accordance with the Declaration of Helsinki, and approved by The Ethic Committee of Peking Union Medical College Hospital (Approval Code: S-478 Approval Date: 2012-10-31).

## Data Availability

The raw blot results are available in the Supplemental Files. The original lectin microarray data is available at EMBL-EBI Biostudies: S-BSST1018.

https://www.ebi.ac.uk/biostudies/studies/S-BSST1018

## Supplemental Information

Supplemental information for this article can be found online at http://dx.doi.org/10.7717/peerj.14853#supplemental-information.

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
