# Peer review of "Retrospective screening of serum IgG glycosylation biomarker for primary Sjögren’s syndrome using lectin microarray"

_PeerJ, doi:10.7717/peerj.14853_

## Round 0.1 · original submission · Minor Revisions

Three experts in this field assessed your manuscript and found the content relevant and well-written. There are some concerns though, which should be addressed in a revised version of the manuscript.

Reviewer 1 ·

Basic reporting

In general, the research is carried out correctly and the methods used to try to answer the research question are adequate.
The manuscript structure is well organized and all raw data is displayed.

Experimental design

The approaches and methods used are correct and well described in the manuscript. Just review a bit the abbreviation of some units, for example, ml--> mL

Although the methods used do not reach the fine point of the issue of glycosylation of antibodies, I consider that it is a good advance for future improvements of the method and thus improve the exact detection of the syndrome.

Validity of the findings

The results shown in the manuscript support the conclusions of the work and although it is missing for the exact detection of the syndrome. The results show important advances.

Additional comments

In the abstract section define: ACL, MNA-M, LCA, SSA, PHA-E and PHA-L.

Line 48: Change the word pathogenesis to disease or illness, but no pathogenesis. The same in line 61.
Line 62: definition of SSB/Ro and SSB/La
Line 131: define HC
Line 229: full name of HC
Line 234: change the word pathogenesis

Reviewer 2 ·

Basic reporting

This work addresses the differences in IgG glycosylation among PSS patients, when compared with healthy controls and PBC controls, differences that are proposed as biomarkers for this autoimmune disease. Lectin microarray analysis were implemented to search for different lectin binding patterns of the antibodies from PSS patients, results that were later confirmed by lectin blot.
The results obtained were satisfactorily analyzed, and may open a new path for the development of a fast and sensitive diagnostic tool for PSS.
The text is well written and explained, with only a few grammatical errors (some marked on the PDF).
The introduction and discussion are very well referenced, providing all the information needed to understand the goal of the paper and to explain the results obtained.
The figures and tables are nicely done and explained, although I have some questions for figures 2 and 3, that I explain in the PDF.

Experimental design

The research question is well defined and important, and answered with the methodology used in the study. However, I do not completely agree with how the lectin blot was performed. The authors explain that the blot was used to validate the microarray results, by comparing the affinity of the lectins to the patients and control antibodies. The total concentration of the patients and control samples used for the blot is not mentioned, making it impossible to know if the signal observed in the blot is due to a higher concentration of the antibodies in the sample or to a higher affinity for the lectins. This should be addressed by the authors. Also, I did not fully understand what samples were analyzed in the lectin blot and how were they selected.

Validity of the findings

The conclusion is concise and summarizes the results, and most importantly, the limitations of the study are mentioned and explained

Annotated reviews are not available for download in order to protect the identity of reviewers who chose to remain anonymous.

Reviewer 3 ·

Basic reporting

The authors present interesting and well written and structured results regarding IgG glycosylation profiles obtained through lectin microarray from patient’s samples with primary Sjogren Syndrome, primary biliary cholangitis, and healthy controls. The introduction requires a clear mention if there is a real referenced need of tools that allow distinguishing PSS from PBC and the clinical situations in which this is required. The results are clear and soundly presented.
Line 66 overlaps/overlapped repeats
Line 73 Change to: IgG is mostly N-glycosylated in the heavy constant region

Experimental design

The experimental design constitutes an original primary research withing the Aims and Scope of the journal. The research question can be better defined in regard to developing tools to distinguish PSS from PBC as this is not very clearly introduced in the manuscript. The description of lectins should indicate the known specificities of each relevant lectin, including the type of linkage and the reference, also stating the full name of each lectin in the methodology. The authors should state why the IgG was not purified and total serum was used.

Validity of the findings

The validity of the findings is sound, nonetheless my main concern it that it is mostly descriptive without further analysis to determine what lectins have diagnostic value to establish a biomarker potential. Is there further statistical analysis that could be performed such as ROC?

Conclusions should be clearer regarding the data and statistical analysis to determine if a lectin or lectin combination is of diagnostic value to distinguish PSS from PBC as it seems that this is the aim of the study and I consider that this can be further supported. Additionally, what was the relation between lectin profiles and autoantibody positivity.

---

## Round 0.2 · accepted · Accept

The new version of the manuscript includes all the points raised by the three Reviewers. As a consequence, the manuscript is now suitable for publication.